# Long-Term Outcome of Infective Endocarditis Involving Cardiac Implantable Electronic Devices: Impact of Comorbidities and Lead Extraction

**DOI:** 10.3390/jcm11247357

**Published:** 2022-12-11

**Authors:** Emanuele Durante-Mangoni, Maria Paola Ursi, Roberto Andini, Irene Mattucci, Ester E. Della Ratta, Domenico Iossa, Lorenzo Bertolino, Stefano De Vivo, Sabrina Manduca, Michele Torella, Marisa De Feo, Rosa Zampino

**Affiliations:** 1Department of Precision Medicine, University of Campania “Luigi Vanvitelli”, Via de Crecchio, 7, 80138 Napoli, Italy; 2AORN Ospedali dei Colli—Ospedale Monaldi, Piazzale Ettore Ruggieri, 80131 Napoli, Italy; 3Department of Translational Medical Sciences, University of Campania “Luigi Vanvitelli”, Via Leonardo Bianchi c/o Ospedale Monaldi, 80131 Napoli, Italy; 4Department of Advanced Medical and Surgical Sciences, University of Campania “Luigi Vanvitelli”, Piazza Luigi Miraglia, 2, 80138 Napoli, Italy

**Keywords:** infective endocarditis, cardiac implantable electronic device, long-term mortality, predictors, transvenous lead extraction

## Abstract

(1) Background: Management of cardiac implantable electronic device-related infective endocarditis (CIED-IE) hinges on complete hardware removal. We assessed whether long-term prognosis is affected by device removal, considering baseline patient comorbid conditions; (2) Methods: A total of 125 consecutive patients hospitalized for CIED-IE were included in this retrospective analysis. Outcomes were in-hospital, one-year, and long-term mortality. There were 109 patients who underwent device removal, 91 by transvenous lead extraction (TLE) and 18 by open heart surgery (OHS); (3) Results: TLE translated into lower hospital mortality (4.4% vs. 22.5% with OHS; *p* = 0.03). Septic pulmonary embolism was the only independent predictor of in-hospital mortality (OR:7.38 [1.49–36.6], *p* = 0.013). One-year mortality was in contrast independently associated to tricuspid valve involvement (*p* = 0.01) and Charlson comorbidity index (CCI, *p* = 0.039), but not the hardware removal modality. After a median follow-up of 41 months, mortality rose to 24%, and was significantly influenced only by CCI. Specifically, patients with a higher CCI who were also treated with TLE showed a survival rate not significantly different from those managed with medical therapy only; (4) Conclusions: In CIED-IE, TLE is the strategy of choice for hardware removal, improving early outcomes. Long-term benefits of TLE are lessened by comorbidities. In cases of CIED-IE with high CCI, a more conservative approach might be an option.

## 1. Introduction

Cardiac implantable electronic device (CIED)-related infective endocarditis (IE) is an emerging clinical condition requiring a complex diagnostic and therapeutic approach [1,2,3]. Successful management hinges on complete device removal [4]. That, however, may not always be feasible. Device removal is usually accomplished by transvenous lead extraction (TLE), although some cases are better managed through open heart surgery (OHS) [5]. In up to 20% of cases [6], device removal in the absence of CIED infection complications is not performed at all, mostly due to high operative risk [6,7,8,9].

Several studies have shown that device removal translates into a survival benefit in the mid-term, i.e., at one year after discharge [10]. However, how this approach affects longer-term prognosis, especially in subjects with a higher burden of comorbidities, is less clear [11,12]. As detailed in the European Heart Rhythm Association (EHRA) international consensus document [13], long-term mortality is 2.5 times higher in patients with CIED infection than in non-infected carriers, and at three-year follow-up, ranges between 14% and 33%. The main factors influencing mortality rates remain unknown and could be related to the CIED infection itself, patients’ comorbidities, or management strategy. Indeed, the long-term outcome of CIED infection has been described by a few studies [14,15,16,17,18] including a limited number of IE patients recruited over a wide timespan. Although most subjects included in these studies underwent device removal, their long-term prognosis seems to be poor nonetheless [12]. More specifically, Deharo et al. [12] showed a mortality rate of 14.3% and 35.3% at 1-year and 3-year follow-ups, respectively, despite the fact that total hardware removal was achieved in 98.5% of cases. Moreover, how the actual therapeutic approach (OHS vs. TLE) impacts long-term outcomes remains unknown.

Stratifying patients according to baseline comorbidities might be of utmost importance in understanding the real benefits of TLE in the long term and identifying a target population that might be unlikely to benefit from that therapeutic strategy. To our knowledge, there are no studies that have been designed to address this point.

Accordingly, in this study we aimed at assessing whether device removal as well as the actual method thereof was related with a better long-term prognosis, taking into account baseline patient comorbid conditions.

## 2. Materials and Methods

This study is a retrospective analysis of outcome data obtained prospectively from consecutive patients hospitalized in our clinical unit between June 2000 and December 2015 because of IE on CIEDs, including pacemakers (PMKs) and automatic implantable cardioverter defibrillators (AICDs).

The study ended in December 2020, allowing for a 5-year follow-up for each patient included. All patients fulfilled diagnostic criteria existing over time (modified Duke criteria [19] adapted to CIED-related IE) for definite IE. All cases were re-evaluated according to currently available EHRA CIED infection criteria [13] and were found to fulfill the diagnosis of definite CIED-IE (2 major criteria or 1 major + 3 minor criteria such as fever or embolic phenomena). Patients with evidence of left heart valve involvement were excluded. No other exclusion criteria were applied. Each patient gave his/her informed consent to participation in the study and data collection at hospital admission. The study observational procedures were approved by the University of Campania “L. Vanvitelli” Ethics Committee and were compliant with the 1976 Declaration of Helsinki and its later amendments (prot. AOC/0011110/2020). All patients gave their informed consent to the anonymous use of their clinical data.

Patients enrolled in the study were all subjects with an age ≥18 years and definite infective endocarditis on PMK or AICD, irrespective of other baseline comorbidities. PMKs were implanted to treat advanced atrio-ventricular blocks, sinus node disease, or brady–tachy syndrome, whilst AICDs were implanted in primary or secondary prevention of sudden cardiac death in patients with cardiomyopathy. To preserve the assumption of independence of observations, only the first episode of infective endocarditis recorded for an individual patient was used in the analysis. Subjects with mere suspicion of CIED-related IE or with definite IE at another location were excluded. A standard case report form was used to collect data, including clinical characteristics (demographics, comorbidities, preexisting cardiac conditions) and details of the IE episode (source of acquisition, microbiology and echocardiography findings, complications, treatment, and outcome).

Patients were managed according to a homogeneous clinical protocol, always including performance of blood cultures and echocardiography studies. A chest CT scan to detect pulmonary embolism was performed in patients with highly elevated D-dimer levels (>1000 ng/dL) or lung consolidations at chest x-ray. We collected and evaluated data on age, sex, date of hospital admission and discharge, interval between first clinical manifestation and diagnosis, initial clinical presentation, and baseline hematochemical parameters.

During most of the study period, no formal “endocarditis team” was in place in our hospital, although each case was managed in the context of a multidisciplinary assessment made by internists, electrophysiologists, echocardiography specialists, and cardiac surgeons. Open heart surgery was performed where additional indications for surgery existed, when vegetations were deemed to be too large for transvenous lead extraction, or in cases this procedure had previously failed elsewhere.

TLE of CIED was always performed in an operating room by a cardiac pacing physician with a cardiac surgeon standby and was followed by intensive care unit monitoring during the initial 24 h to check for immediate complications. In pacing-dependent patients, a temporary transvenous lead was left in place until device reimplantation. TLE was performed in our center at the time of this study using standard locking stylets and either mechanical or Evolution sheaths, but without laser-powered sheaths. None of the included cases underwent percutaneous aspiration. Most patients underwent TLE after several days of antibiotic therapy (first empiric, then targeted), when bacteremia was over, sepsis had waned, and the patient had become stable, in order to lower the risk of septic pulmonary embolism and periprocedural complications [3]. This strategy also allowed us to minimize temporary pacing and anticipate device reimplantation. Patients who underwent OHS had an epicardial stimulator implanted. The actual device-removal strategy was chosen by our multidisciplinary team on a case-by-case basis, considering patient clinical conditions, degree of cardiac dysfunction, presence of additional indications for OHS (valve replacement, coronary revascularization), number of leads, size of vegetations, and previous failed attempts at TLE. None of the patients included in this study underwent percutaneous vegetation extraction.

After discharge, all patients entered an active follow-up by means of outpatient clinic assessments, blood tests, and subsequent phone interviews.

The following comorbidities were assessed: prior or de novo congestive heart failure (CHF); chronic obstructive pulmonary disease (COPD); chronic kidney disease (CKD); diabetes mellitus (DM); cancer; liver disease; coronary, cerebral, or peripheral arterial disease; dialysis; dementia; and illicit drug use. Age-adjusted Charlson comorbidity index (CCI) was computed in light of its predictive validity for a number of clinical outcomes.

Data regarding therapy included antimicrobial agents administered and surgical treatment and the modality thereof. Major covariates we considered as possibly influencing outcome were *Staphylococcus aureus* infection, AICD vs. PMK infection, tricuspid valve involvement, vegetation size, pulmonary embolism, treatment modality, and CCI. Outcomes of interest were in-hospital, one-year, and longer-term mortality.

Differences between groups were computed through the Fisher’s exact test for categorical variables and the Mann–Whitney U test or Kruskal–Wallis test for continuous variables. Odds ratios (ORs) with 95% confidence intervals (CIs) were estimated through logistic regression with covariates extracted among variables significantly associated with the outcome of interest on univariate analysis. Overall survival was estimated using the Kaplan–Meier method [6]. Significance of observed differences was assessed by the Mantel–Cox log-rank test. Follow-up duration was calculated from the date of admission to the date of death or the last available follow-up. To take into account the influence of comorbidities on outcome across groups managed with TLE vs. medical therapy only, a CCI-matched subgroup of TLE subjects was identified and compared to subjects managed with medical therapy only.

Numerical data are shown as mean ± standard deviation or median with range, whilst categorical variables and qualitative data are presented as number and percentage.

All analyses were performed using the Windows statistical software SPSS 22 (SPSS, Inc., Chicago, IL, USA).

## 3. Results

### 3.1. Clinical Characteristics of the Study Cohort

From June 2000 to December 2015, we observed 125 cases of CIED-related endocarditis fulfilling the inclusion criteria. Their median age was 67 yrs (range 17–90), and most were males (80%).

Infection involved PMK leads in 85 cases (68%), AICD leads in 40 (32%), and extended to tricuspid valve in 16 cases (12.8%). Based on the presumed pattern of infection acquisition [5], 80% of cases were deemed to be nosocomial, 4% non-nosocomial healthcare-related, and 16% community-acquired or spontaneous. The mean time from first clinical manifestation of infection to the diagnosis of CIED-related IE was less than one month in 49 cases (39.2%), between 1 and 6 months in 55 (44%), and above 6 months in 21 (16.8%).

Clinical and hematochemical parameters of CIED-related IE patients on admission are shown in Table 1. Inflammatory markers were overall weakly elevated, and a subset of patients had neither C-reactive protein nor erythrocyte sedimentation rate elevation. Likewise, fever was absent in 12.8% of cases. Pulmonary embolic events occurred before diagnosis in 28.8% of cases. Overall, there was a high rate of prior CHF, COPD, DM, and CKD (Table 1).

### 3.2. Microbiological Findings

Microbiological data are summarized in Table 2. Blood cultures were positive in 92 cases (79.3%). Additional cultures were performed in 106 patients (including the endovascular lead in 87 cases and the excised tricuspid valves in 3), and specifically in 31 of the 33 cases with negative blood cultures.

The most commonly isolated pathogens were staphylococci, 53 of which were methicillin-resistant (MR, 54.1%). Specifically, MR was expressed by 18.9% of *S. aureus* and 81.1% of coagulase-negative staphylococci (CoNS).

### 3.3. Echocardiography Data

Transthoracic echocardiography was performed in all patients and a transesophageal study in 85 of them (68%). Lead vegetations were found in 94 patients (75.2%) and involvement of the tricuspid valve with presence of vegetations and valvular regurgitation was observed in 16 patients (12.8%).

Median vegetation size was 17 mm (range: 3–40 mm). Vegetation size was ≤10 mm in 26 cases (32.1%).

### 3.4. Therapeutic Approach

All CIED-related IE cases were treated with antibiotics, for a median of 42 days (range: 2–150 days). Treatment duration was 42 [42–54] days in TLE and OHS groups (surgical group) vs. 42 [35–49] days in the medical therapy only group (*p* = 0.216). The most commonly used molecules were daptomycin (61 cases; 48.8%), amoxicillin-clavulanic acid (34 cases; 27.2%), and glycopeptides (21 cases; 16.8%).

A total of 109 subjects (87.2%) underwent surgical removal of the entire device. In details, 18 patients (16.5%) underwent open heart surgery (OHS), with complete extraction of all hardware as well as removal of residual vegetations from the tricuspid valve or valve replacement. The remaining 91 patients (83.5%) underwent transvenous lead extraction (TLE) and generator removal. A new device was re-implanted in 103 patients (94.5%), a median of 4 days after surgery. Re-implantation was performed only when blood cultures drawn the day of device removal were negative for at least 48 h. In 6 cases, there was no indication for CIED reimplantation. The median duration of pre-operative antimicrobial therapy was 16 days. Antimicrobials were administered after hardware removal for a median of 24 days.

### 3.5. Complications and Short-Term Outcome

The most common complications were septic pulmonary embolism (observed overall, before and after lead extraction, in 48 cases [38.4%]), acute or acute-on-chronic heart failure (25 cases [20%], of whom 9 were in NYHA class IV), and subclavian or axillary venous thrombosis (5 cases). Persistence of the fibrous sleeve enveloping the electrocatheter after TLE, also known as a “ghost”, occurred in 7 cases (7.6% of TLE cases). There was 1 case of Takotsubo cardiomyopathy and 2 cases of post-procedural acute coronary syndrome. Worsening tricuspid valve regurgitation after TLE occurred in 1 case. No early relapse of infection occurred, although in 3 cases (2.4%) CIED infection recurred during follow-up. There were no cases of heart rupture, pericardial effusion, or superior vena cava tear. In two patients, post-procedural pneumothorax occurred.

Overall, in-hospital mortality was 8%. Predictors of hospitalization outcome are shown in Table 3. Occurrence of pulmonary embolism was the only independent predictor of hospital mortality (OR 7.5 [95% C.I. 1.5–37]; *p* = 0.013). This finding was not due to a higher rate of CT scan performance in patients deteriorating and subsequently dying. Although a clear association between chest CT performance and diagnosis of septic pulmonary embolism was evident (data not shown), patients who died did not undergo chest CT scan more frequently than those who survived (57% vs. 69%; *p* = 0.676). In the TLE group, 25 patients (27%) had pulmonary embolism before the procedure. This complication occurred as a direct consequence of TLE in 37 patients (40%).

Mortality was not affected by the type of infection acquisition, the absence of fever or inflammatory marker elevation, or diagnostic delay (data not shown).

The therapeutic approach (medical, OHS, and TLE) was associated with the outcome of hospitalization on the univariate analysis. In particular, death occurred in 2 of 16 subjects treated with medical therapy only (12.5%), 4 of 18 underwent OHS (22.2%), and 4 of 91 managed with TLE (4.4%) (*p* = 0.03). Table 4 shows clinical features of patients grouped according to the therapeutic approach. Those treated with medical therapy only were significantly older than surgical cases, both OHS and TLE (*p* = 0.01 and *p* = 0.009, respectively), and showed a significantly higher CCI (*p* = 0.017). The lowest CCI was observed among OHS patients, who also had a higher rate of tricuspid valve involvement (*p* = 0.002) (Table 4). After matching patients for CCI, the therapeutic approach (medical therapy only vs. TLE) was no longer associated with the outcome of hospitalization. Specifically, 55 TLE patients with a median CCI of 6 (range 0–14) showed a hospital mortality that was not significantly different from the 16 medically treated patients (dead at discharge: 3 [5.5%] and 2 [12%], respectively; *p* = 0.58).

**Table 3 jcm-11-07357-t003:** Univariate and multivariate analysis of factors associated with in-hospital mortality.

	In-Hospital Mortality (Univariate)			In-Hospital Mortality (Multivariate)	
	**Alive (n = 115)**	**Dead (n = 10)**	** *p* **	**OR ^1^ (95% C.I.)**	** *p* **
Age, mean (SD ^2^)	67 (17–90)	68 (40–82)	0.884		
Sex, n (%)			1.000		
*Male*	92 (80)	8 (80)			
*Female*	23 (20)	2 (20)			
CIED ^3^ type, n (%)			0.499		
*PMK* ^4^	77 (67)	8 (80)			
*AICD* ^5^	38 (33)	2 (20)			
*S. aureus* infection, n (%)	32 (27.8)	1 (10)	0.289		
Tricuspid valve involvement	13 (11.3)	3 (30)	0.118		
Vegetation size, n (%)			0.341		
*<10 mm*	65 (56.5)	4 (40)			
*≥* *10 mm*	50 (43.5)	6 (60)			
Pulmonary embolism, n (%)	40 (34.8)	8 (80)	0.013	7.388 [1.489–36.664]	0.014
CCI ^6^, median (range)	5 (0–14)	6 (3–14)	0.968		
Therapeutic approach			0.030	1.991 [0.559–7.095]	0.288
*Medical*	14 (12.2)	2 (20)			
*OHS* ^7^	14 (12.2)	4 (40)			
*TLE* ^8^	87 (75.6)	4 (40)			

^1^ OR = odds ratio; ^2^ SD = standard deviation; ^3^ CIED = cardiac implantable electronic device; ^4^ PMK = pacemaker; ^5^ AICD = automated implantable cardioverter defibrillator; ^6^ CCI = Charlson comorbidity index; ^7^ OHS = open heart surgery; ^8^ TLE = transvenous lead extraction.

Predictors of one-year mortality are shown in Table 5. Tricuspid involvement and CCI were the variables independently associated with this outcome (*p* = 0.010 and *p* = 0.039, respectively).

In contrast, one-year survival did not differ significantly as a result of the therapeutic approach (*p* = 0.160) (Figure 1).

### 3.6. Long-Term Outcome

At the last follow-up available, after a median observation time of 41 months (range: 1–146 months), mortality rose to 24%. Late death occurred in a further 5 cases of the medical therapy group (35.7%), in 1 of those subjected to OHS (7.1%), and in 14 patients treated with TLE (16.1%). The substantial increase in mortality during follow-up suggested an effect of other clinical variables beyond the actual therapeutic strategy used. Further analysis of predictors of late mortality showed that CCI was significantly associated with this outcome (*p* = 0.049) (data not shown). In particular, a CCI higher than the group median (≥6 for TLE patients and ≥5 for patients treated with medical therapy only) translated into a significantly lower likelihood of survival (Figure 2). However, within the subgroup of patients with higher CCI, survival was not significantly different when comparing patients who underwent TLE with those managed with medical therapy only (Figure 3). Reasons for death during long-term outcomes were not available for most patients.

## 4. Discussion

Based on our data, TLE may not translate into a substantial survival benefit in patients with advanced age and a high burden of comorbidities, i.e., with a high CCI. This should be taken into account in light of the peri-procedural risks of TLE as compared to the often subacute course of CIED-related IE. Undoubtedly, these results need confirmation on large prospective cohorts, but may help clinicians in handling the choice of device removal in this setting. Moreover, the results of our study are in line with recent literature showing that age and comorbidities are among the major drivers of both short- and long-term outcomes of CIED-IE [20,21,22]. A composite measure of aging and multi-morbidity [23], the Charlson comorbidity index appeared to be the strongest and most consistent factor associated with in-hospital, one-year, and long-term mortality. In contrast, our experience suggests that the therapeutic approach (medical vs. OHS vs. TLE) is of lesser importance when variably applied to different patient subgroups. 

We studied a large sample of patients from a single center, managed with a standardized approach. This cohort was inclusive of both PMK and AICD carriers, with staphylococci being the largely predominant etiology and a range of presentations spanning from mild to very severe infection. Therefore, we believe our experience may have fair generalizability. Indeed, although allocation to each therapeutic approach was not randomized, we replicated previous observations [24,25] showing an increased hospital mortality with both OHS (22.5%) and medical therapy (12.5%) compared to TLE (4.4%). It is interesting to note that patients treated with OHS were characterized by significantly lower CCI values and experienced mostly early, procedure-related mortality. Analysis of long-term outcomes after TLE showed that after the initial survival advantage of this therapeutic approach, a substantial mortality is nonetheless observed in patients with a high burden of comorbidities coupled with an advanced age. This finding corroborates the results emerging from previous observations [12].

Our study has some clear limitations. It was a single-center study in a setting where specific expertise was acquired through referral of patients from local health care facilities. Allocation to the different therapeutic approaches was not randomized and took into account several factors. Patients who refused to undergo device removal received only medical treatment. Open heart surgery was an option for patients with preserved ejection fraction, with larger vegetations (>2 cm) on device leads, and additional surgical indications, including the need for coronary artery bypass grafting or tricuspid valve repair. All other patients underwent transvenous lead extraction. Random allocation could not be performed as the accepted standard strategy for device removal has been TLE since about 2004. As a result, there were some significant baseline differences among groups, which have been described in detail in Table 4. Despite having fewer comorbidities, fewer defibrillators, and less frequent *S. aureus* etiology, patients who underwent OHS showed a significantly higher in-hospital mortality. Thus, a randomized, prospective study would remain fundamental to determining whether the therapeutic strategy independently affects the outcome of CIED-related IE.

## 5. Conclusions

In conclusion, this study suggests that the therapeutic approach to CIED-related IE should take into account patient age and comorbidities. TLE confirms the strategy of choice for hardware removal in terms of early and mid-term outcome and should always be pursued whenever feasible. In the not-infrequent cases with high CCI, a more conservative approach might not necessarily be disregarded. OHS should be reserved to cases of TLE failure when no significant comorbidities exist.

## Figures and Tables

**Figure 1 jcm-11-07357-f001:**
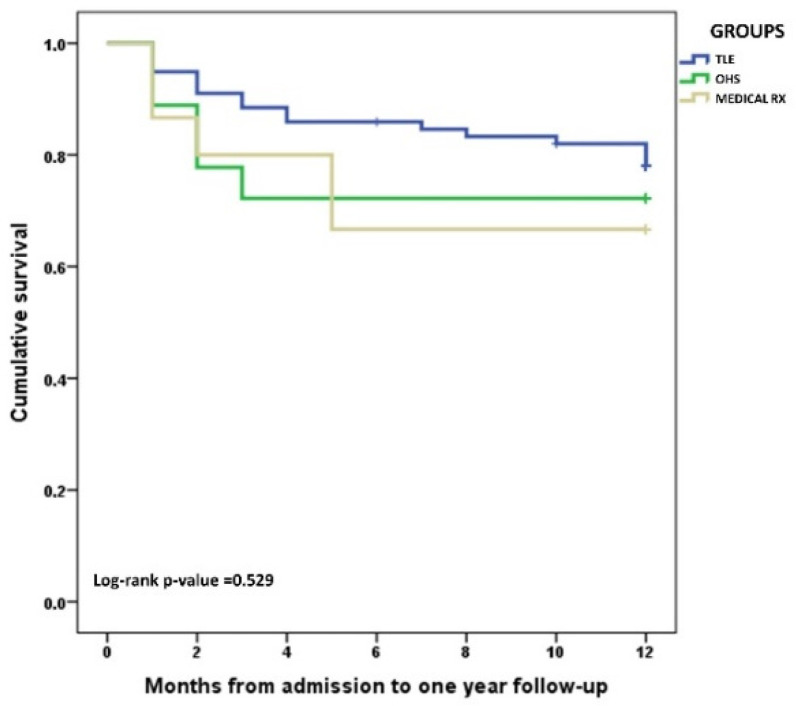
Kaplan–Meier curves describing survival probability, after one year of follow-up, according to the three therapeutic approaches used for CIED-related endocarditis: medical only, transvenous lead extraction (TLE), and open heart surgery (OHS).

**Figure 2 jcm-11-07357-f002:**
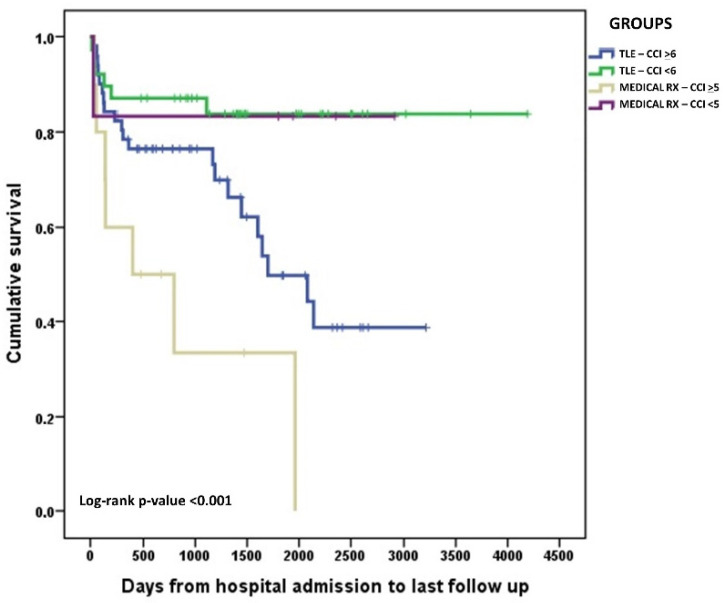
Kaplan–Meier curves describing long-term survival probability according to therapeutic approach (medical only vs. TLE) and CCI < or ≥ the group median.

**Figure 3 jcm-11-07357-f003:**
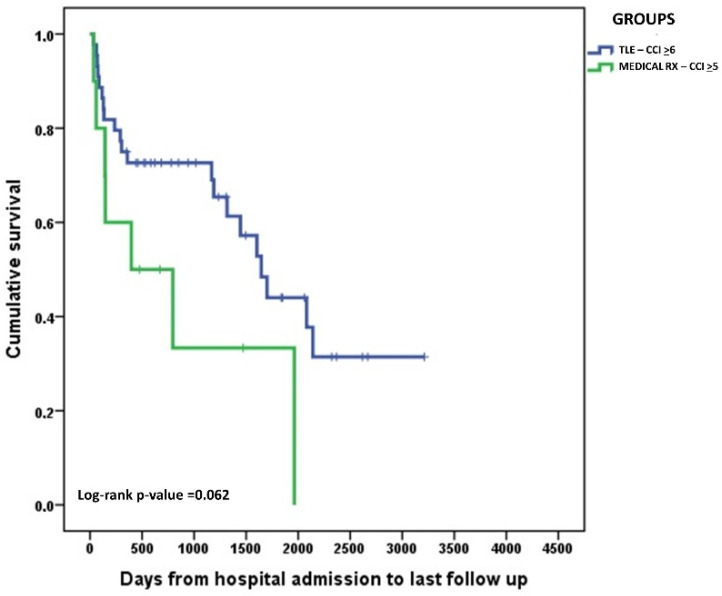
Kaplan–Meier curves comparing long-term survival probability of the two subgroups of patients with a CCI higher that the group median managed with medical therapy only vs. TLE.

**Table 1 jcm-11-07357-t001:** Clinical and hematochemical parameters of CIED-related IE patients on admission (n = 125).

Parameter	Number (%)	Median [Range]
Age (yrs)		67 [17–90]
Males	100 (80)	
Vegetation size (mm)		17 [3–40]
Elevated CRP	106 (84.8)	
CRP, mg/dL		3.3 [0.1–34.2]
Elevated ESR	97 (88.2)	
ESR, (mm)		42 [4–120]
White blood cells (WBC)		8840 [1048–12,810]
Platelets (PLT)		178,000 [31,000–375,000]
Hemoglobin (HGB) g/dL		11.8 [5.4–16.1]
D-dimers, ng/mL		545 [45–44,450]
INR		1.24 [0.8–7.69]
Creatinine, mg/dL		1.1 [0.1–10.8]
Glucose, mg/dL		107 [59–496]
Troponin, mg/dL		0.03 [0–23.19]
**Clinical Manifestations at Onset**		
Fever	109 (87.2)	
Splenomegaly	23 (18.4)	
Embolic events	51 (40.8)	
New onset cardiac murmur	17 (13.6)	
Worsening of pre-existing murmur	18 (14.4)	
Hematuria	19 (15.2)	
**Comorbidities**		
Charlson comorbidity index		5 [0–14]
Chronic obstructive pulmonary disease	32 (25.6)	
Prior acute myocardial infarction	37 (29.6)	
Chronic heart failure	60 (48)	
Peripheral arterial disease	12 (9.6)	
Cerebrovascular disease	12 (9.6)	
Diabetes mellitus	36 (28.8)	
Chronic kidney disease (CKD)	35 (28)	
CKD on hemodialysis	6 (4.8)	
Intravenous drug use	1 (0.8)	
Cancer	10 (8)	
Chronic liver disease	13 (10.4)	
Previous endocarditis	4 (3.2)	

**Table 2 jcm-11-07357-t002:** Microbiological isolates of CIED-related IE patients.

Pathogen	Number (%)
Staphylococci	98 (78.4)
*Staphylococcus aureus*	31 (31.6)
*Staphylococcus epidermidis*	49 (50)
*Non-epidermidis coagulase-negative staphylococci*	18 (18.4)
Enterococcus faecalis	2 (1.7)
Viridans group Streptococci	2 (1.7)
*Peptostreptococcus* spp.	2 (1.7)
Propionibacterium acnes	2 (1.7)
Pseudomonas aeruginosa	3 (2.4)
*Candida* spp.	2 (1.7)
Gram-negative bacilli ^1^	6 (4.8)
Negative cultures	8 (6.4)

^1^
*E. coli, S. marcescens, A. baumannii, E. cloacae.*

**Table 4 jcm-11-07357-t004:** Clinical features of patients grouped according to the actual therapeutic approach.

	Medical Therapy	Open Heart Surgery (OHS ^1^)	Transvenous Lead Extraction (TLE ^2^)	*p*
**N**	**16**	**18**	**91**	
**Years enrollment**	**2001–2015**	**2000–2012**	**2004–2015**	
Age, mean (SD ^3^)	74.1 (10.4)	63.4 (15.6)	61.3 (15.5)	0.378
Sex, n (%)				0.532
*Male*	12 (75)	13 (72.2)	75 (82.4)	
*Female*	4 (25)	5 (27.8)	16 (17.6)	
CIED ^4^, n (%)				0.019
*PMK* ^5^	12 (75)	17 (94.4)	56 (61.5)	0.076
*AICD* ^6^	4 (25)	1 (5.6)	35 (38.5)	0.002
*S. aureus*, n (%)	2 (12.5)	2 (11.1)	29 (31.9)	0.121
Tricuspid involvement	4 (25)	6 (33.3)	6 (6.6)	0.850
Vegetation size, n (%)				0.017
*<10 mm*	11 (68.7)	12 (66.7)	44 (48.9)	0.030
*≥* *10 mm*	5 (31.3)	5 (33.3)	46 (51.1)	
Pulmonary embolism, n (%)	6 (12.5)	8 (16.7)	34 (70.8)	0.898
CCI ^7^, median (range)	6 (4–10)	4 (0–10)	5 (0–14)	
In-hospital mortality, n (%)	2 (12.5)	4 (22.5)	4 (4.4)	0.006
Coronary artery disease, n (%)				0.378
Yes/No	1 (6.7)/14 (93.3)	2 (11.1)/16 (89.9)	9 (10.2)/79 (89.8)	
Heart failure, n (%)				0.532
Yes/No	6 (7.5)/10 (62.5)	3 (16.7)/15 (83.3)	51(56)/40 (44)	

^1^ OHS = open heart surgery; ^2^ TLE = transvenous lead extraction.^3^ SD = standard deviation; ^4^ CIED = cardiac implantable electronic device; ^5^ PMK = pacemaker; ^6^ AICD = automated implantable cardioverter defibrillator; ^7^ CCI = Charlson comorbidity index.

**Table 5 jcm-11-07357-t005:** Univariate and multivariate analysis of one-year mortality predictors.

	One-Year Mortality (Univariate)			One-Year Mortality (Multivariate)	
	**Alive (n = 104)**	**Dead (n = 11)**	** *p* **	**OR ^1^ (95% C.I.)**	** *p* **
Age, mean (SD ^2^)	66 (17–90)	75 (54–85)	0.034	1.052 [0.973–1.137]	0.202
Sex, n (%)			1.000		
*Male*	83 (79.8)	9 (81.8)			
*Female*	21 (20.2)	2 (18.2)			
CIED ^3^ type, n (%)			1.000		
*PMK* ^4^	70 (67.3)	7 (63.6)			
*AICD* ^5^	34 (32.7)	4 (36.4)			
*S. aureus* infection, n (%)	28 (26.9)	4 (36.4)	0.496		
Tricuspid valve involvement	9 (8.7)	4 (36.4)	0.021	8.240 [1.667–40.736]	0.010
Vegetation size, n (%)			0.658		
*<10 mm*	22 (32.8)	1 (16.7)			
*≥* *10 mm*	45 (67.2)	5 (83.3)			
Pulmonary embolism, n (%)	37 (35.6)	3 (27.3)	0.745		
CCI ^6^, median (range)	5 (0–14)	7 (4–13)	0.025	1.325 [1.105–1.730] *	0.039
Therapeutic approach			0.791		
*Medical*	12 (11.5)	2 (18.2)			
*OHS* ^7^	13 (12.5)	1 (9.1)			
*TLE* ^8^	79 (76)	8 (72.7)			

^1^ OR = odds ratio; ^2^ SD = standard deviation; ^3^ CIED = cardiac implantable electronic device; ^4^ PMK = pacemaker; ^5^ AICD = automated implantable cardioverter defibrillator; ^6^ CCI = Charlson comorbidity index; ^7^ OHS = open heart surgery; ^8^ TLE = transvenous lead extraction; * per 1 point increase in CCI.

## Data Availability

Data are made available by the corresponding author upon reasonable request.

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
