# Peer review of "Long-Term Outcome of Infective Endocarditis Involving Cardiac Implantable Electronic Devices: Impact of Comorbidities and Lead Extraction"

_jcm, 2022, doi:10.3390/jcm11247357_

Round 1

Reviewer 1 Report

The authors make effort in CIED-related endocarditis, which showed TLE is relatively safe and efficient management in low CCI patients. Pulmonary embolism is a significant predictor of in-hospital mortality.

One error needs to be clarified. Table 5, one-year mortality, but the column headline "in-hospital mortality"?

I have some comments to the authors.

First, for pulmonary embolism is the only predictor of in-hospital mortality in multivariable analysis, authors need to explain their strategy in arrange CT scan. 57% of expired patients and 69% of survived patients received chest CT scan, why? Did these patients have any symptoms of pulmonary embolism? When did these patients receive CT scan? Before or after TLE? Did TLE cause iatrogenic pulmonary embolism?

Second, in the predictor analysis of mortality (table 3), the therapeutic approach should not be a factor. The choice of therapeutic approach in this study is not randomized. Physicians chose medical/ OHS/ TLE according to patients' condition and IE severity. The therapeutic approach is a result after evaluating patients. Therefore, it cannot be a predictor of mortality.

Author Response

Reply to Reviewer 1

We would like to thank the Reviewer for the valuable comments and suggestions on our manuscript. We did our best to incorporate them, preparing a revised draft, which appears to us significantly improved.

Below, please find a detailed point-by-point reply to each comment.

The authors make effort in CIED-related endocarditis, which showed TLE is relatively safe and efficient management in low CCI patients. Pulmonary embolism is a significant predictor of in-hospital mortality.

One error needs to be clarified. Table 5, one-year mortality, but the column headline "in-hospital mortality"?

Reply: We have amended this typing error. Sorry.

I have some comments to the authors.

First, for pulmonary embolism is the only predictor of in-hospital mortality in multivariable analysis, authors need to explain their strategy in arrange CT scan. 57% of expired patients and 69% of survived patients received chest CT scan, why? Did these patients have any symptoms of pulmonary embolism? When did these patients receive CT scan? Before or after TLE? Did TLE cause iatrogenic pulmonary embolism?

Reply: We generally ordered a chest CT scan when D-dimer levels were highly elevated or elevated out of proportion to the other inflammatory markers. Also, we did chest CT scan in patients who showed lung consolidations at chest x-ray. Symptoms of pulmonary embolism were rarely specific and were very difficult to differentiate from symptoms of infection. However, it should be considered that not all patients were in clinical conditions allowing safe performance of chest CT scan. Therefore, the different rates of chest CT scan performance between deceased and survived patients. Regarding timing of chest CT scan, most patient did it before TLE and very few did it after TLE when concurrent hemorrhagic complications made anti-coagulant therapy less feasible. We have now specified our policy for chest CT scan in Methods, lines 101-103, and have entered data on timing of pulmonary embolism in Results, lines 226-228.

Second, in the predictor analysis of mortality (table 3), the therapeutic approach should not be a factor. The choice of therapeutic approach in this study is not randomized. Physicians chose medical/ OHS/ TLE according to patients' condition and IE severity. The therapeutic approach is a result after evaluating patients. Therefore, it cannot be a predictor of mortality.

Reply: we thank the Reviewer for this important comment. However, we have to underscore the primary aim of this study was to assess how the therapeutic approach (OHS vs TLE) impacted long term outcome of CIED-related IE. This cannot be done if the therapeutic approach is not considered among factors possibly affecting mortality, also in-hospital mortality. We agree that the therapeutic approach was not randomly chosen, but this is not a pre-requisite to assess its influence on outcome in clinical studies. It is true that the therapeutic approach was a result after evaluating patients, but this treatment was decided before any death occurred. Thus, we strongly believe that the therapeutic approach should remain among variables in the univariate and multivariable analysis of mortality.

Reviewer 2 Report

This manuscript represented us how we treat patients with implantable electronic device-related infective endocarditis (CIED-IE). Because CIEDs are increasing as increase in aging population and as increase in indications for device implantation (for example, implantable cardioverter defibrillator for primary prevention of sudden cardiac death in patients with decreased cardiac function), these knowledges are very important.

There are some concerns as follows.

1, Table 5 showed one-year mortality. But in the first line, two sentences of “In-hospital mortality” are seen. These may be simple mistake of “one-year mortality”.

2, About antibiotic treatment. Is there any difference in duration of antibiotic medication between surgical (OHS and TLE) and medical treatment?

3, Do authors have any data about causes of death during long follow up? Figure 2 shows obvious prognostic difference according to CCI. Causes of death might be different among those groups. If you have any data, please put some comments about the differences in causes of death among groups.

Author Response

Reply to Reviewer 2

We would like to thank the Reviewer for the valuable comments and suggestions on our manuscript. We did our best to incorporate them, preparing a revised draft, which appears to us significantly improved.

Below, please find a detailed point-by-point reply to each comment.

This manuscript represented us how we treat patients with implantable electronic device-related infective endocarditis (CIED-IE). Because CIEDs are increasing as increase in aging population and as increase in indications for device implantation (for example, implantable cardioverter defibrillator for primary prevention of sudden cardiac death in patients with decreased cardiac function), these knowledges are very important.

There are some concerns as follows.

1, Table 5 showed one-year mortality. But in the first line, two sentences of “In-hospital mortality” are seen. These may be simple mistake of “one-year mortality”.

Reply: Yes, we did a mistake and we have amended this typing error.

2, About antibiotic treatment. Is there any difference in duration of antibiotic medication between surgical (OHS and TLE) and medical treatment?

Reply: No, there was no difference. Duration of treatment did not vary according to the therapeutic approach chosen. In details the median duration of antimicrobial treatment was 42 [42-54] days in TLE and OHS groups (surgical group) vs 42 [35-49] days in the medical therapy only group (p=0.216). We have now added this data in Results, lines 195-196.

3, Do authors have any data about causes of death during long follow up? Figure 2 shows obvious prognostic difference according to CCI. Causes of death might be different among those groups. If you have any data, please put some comments about the differences in causes of death among groups.

Reply: Unfortunately, we have data on the reasons for death during long term outcome only in a few patients, such that we are unable to provide further data on this issue. We have specified this in the manuscript, Results, lines 277-278.